# Emerging Electrochemical Sensors for Real-Time Detection of Tetracyclines in Milk

**DOI:** 10.3390/bios11070232

**Published:** 2021-07-09

**Authors:** Magdalena R. Raykova, Damion K. Corrigan, Morag Holdsworth, Fiona L. Henriquez, Andrew C. Ward

**Affiliations:** 1Civil and Environmental Engineering, University of Strathclyde, Glasgow G1 1XJ, UK; magdalena.raykova@strath.ac.uk; 2Biomedical Engineering, University of Strathclyde, Glasgow G1 1QE, UK; damion.corrigan@strath.ac.uk; 3Graham’s Dairy Family, Bridge of Allan, Stirling, Glasgow FK9 4RW, UK; morag.holdsworth@grahamsfamilydairy.com; 4School of Health and Life Sciences, University of the West of Scotland, Paisley PA1 2BE, UK; fiona.henriquez@uws.ac.uk

**Keywords:** antimicrobial resistance, tetracyclines, dairy, milk, electrochemical biosensors, antimicrobial residues

## Abstract

Antimicrobial drug residues in food are strictly controlled and monitored by national laws in most territories. Tetracyclines are a major broad-spectrum antibiotic class, active against a wide range of Gram-positive and Gram-negative bacteria, and they are the leading choice for the treatment of many conditions in veterinary medicine in recent years. In dairy farms, milk from cows being treated with antibiotic drugs, such as tetracyclines, is considered unfit for human consumption. Contamination of the farm bulk tank with milk containing these residues presents a threat to confidence of supply and results in financial losses to farmers and dairy. Real-time monitoring of milk production for antimicrobial residues could reduce this risk and help to minimise the release of residues into the environment where they can cause reservoirs of antimicrobial resistance. In this article, we review the existing literature for the detection of tetracyclines in cow’s milk. Firstly, the complex nature of the milk matrix is described, and the test strategies in commercial use are outlined. Following this, emerging biosensors in the low-cost biosensors field are contrasted against each other, focusing upon electrochemical biosensors. Existing commercial tests that identify antimicrobial residues within milk are largely limited to beta-lactam detection, or non-specific detection of microbial inhibition, with tests specific to tetracycline residues less prevalent. Herein, we review a number of emerging electrochemical biosensor detection strategies for tetracyclines, which have the potential to close this gap and address the industry challenges associated with existing tests.

## 1. Introduction

Since their discovery and exploitation over 100 years ago, antimicrobial drugs have become an essential part of human and veterinary medicine [1]. Hundreds of individual drugs have been identified from natural sources or synthesised. Some examples of antibiotic classes include β-lactams (e.g., benzylpenicillin), tetracyclines, macrolides and aminoglycosides. Excessive use and exposure of antibiotics produce a global issue with a negative impact on their effectiveness, termed antimicrobial resistance (AMR) [2]. The emergence of AMR is caused by the use of antimicrobial drugs in veterinary practice as well as human medicine. In particular, in farming, apart from treating bacterial infections in food-producing animals, such as cattle, antimicrobials have been used for prophylactics and growth promotion [3]. This excessive use has been recognised as an unnecessary overuse in the European Union (EU), and although antibiotics for growth promotion have been banned, their prophylactic use is still legal [4].

The dairy industry represents a significant proportion of agricultural productivity in many countries globally. For example, within the UK, in 2018, dairy represented 16.9% of the agricultural output and had a market value of £4.5 billion [5]. Milk is used to produce a wide range of foods, such as yoghurt, cheese, butter and cream. If antimicrobial residues are present within milk, there is a negative effect on the processes that rely on microorganisms for production, such as yoghurt and cheese [6,7].

In addition, the presence of antimicrobial residues can have deleterious health impacts, such as the evolution of AMR within the gut microbiome and allergic reactions in sensitive individual consumers [8,9,10]. Therefore, to prevent this, authorities place legal limits on the concentrations of drug residues permitted. In the EU, Maximum Residue Limits (MRLs) are specified for all antimicrobial drugs indicated for veterinary use [3,11]. Within the EU, for example, these are defined in µg antibiotic per kg food product [12]. 

In the context of dairy farming, a cow eliminates antibiotics through the liver, kidneys and udder and many are excreted in an unchanged and active form via all three excretion routes [3]. Depending on drug solubility, antimicrobials are excreted by mammary gland in the range of 0.002–6.0% as the percentage recovery of administrated dose [13,14]. Given that antimicrobial residues can be present in milk from cows treated with antimicrobial drugs, it must be excluded from the milk supply chain to comply with regulations, protect human health and prevent downstream production processes. To support this, minimum withdrawal periods have been defined for each drug between the last administration and reintroduction of milk to the supply chain [15]. Despite this, traces of antibiotics are sometimes present in milk even after the withdrawal period [3]. Moreover, some antibiotic classes are found to be stable in a wide range of temperatures and are stable in milk even after a pasteurisation process reaching temperatures up to 100 °C [10]. 

Tetracyclines were discovered more than half a century ago [16], and the sale of tetracyclines has increased to the extent of becoming the leading choice of antibiotics used on farm animals and cows, in particular, within European countries. The general sale of antibiotics for use on food-producing animal species in the UK has been reported steady over the period between 2014 and 2019. In 2019 alone, 232.2 tonnes of active ingredients were sold in the category of farm animals with 12 tonnes for use on cattle only [17]. Tetracyclines are reported to be the most sold antibiotic class, with 32% of the total sale in 2019. Figure 1 is a comparison of the sale between tetracyclines and the second most sold antibiotic class—beta-lactams—between 2015 and 2019. 

In addition to the legal implications of antimicrobial residues in milk, safe and effective disposal of the milk is also problematic. Two disposal routes have been set by the Food Standards Agency to dispose of contaminated milk. These are: discarding it in a slurry tank or spraying it over the land on the farm [18]. Both routes are a direct exposure of antimicrobials to the environment. Active and unchanged antibiotic molecules or their metabolites can then persist in the environment for long periods of time, driving antimicrobial resistance that can be passed onto farm animals and humans [8,19]. This makes the future treatment of bacterial infections more complicated to almost impossible in some cases [20]. Moreover, antimicrobial resistance is declared one of the top ten global public health threats facing humanity by the World Health Organisation (WHO) [21,22]. The ability to rapidly identify contaminated milk at the source provides more opportunities to dispose of milk economically. Furthermore, developing a greater understanding of the extent and persistence of antimicrobials from farming within the environment is essential to the development of focused mitigation strategies.

Biosensors are devices that have the capability to measure biological or chemical reactions in proportion to the concentration of a target analyte [23]. This technology has been widely employed in healthcare for monitoring health conditions, such as glucose levels in the blood (diabetes) or the presence of human chorionic gonadotropin (HCG) hormone in urine for pregnancy testing [24]. Biosensors can be developed to be rapid, low cost, easy to use and with little to no requirement for laboratory infrastructure. These properties make them attractive for use for farm monitoring and detection of antimicrobial residues in milk.

Real-time sensors for the detection of antimicrobial residues would be of great value at a number of different points in the milk supply chain. Several recent reviews focus upon the development of beta-lactam biosensors [25,26,27,28]. However, to the best of our knowledge, no up-to-date review exists on biosensors for the detection of tetracycline residues in milk. We address this gap here by providing a review of the latest literature on electrochemical sensors for real-time, low-cost detection of tetracycline residues in milk. We describe the important properties of the milk matrix and tetracycline drug class that are important from the perspective of biosensor development before going on to explaining the current techniques that are used to detect residues. This sets the context for the final part of our review, where we focus upon the latest research into electrochemical biosensors to detect tetracycline drugs. Non-electrochemical sensor approaches, such as optical and piezoelectric detection methods, are outside the scope of this review. For a comprehensive review on optical detection, see Chen and Wang, 2020 [29].

## 2. Cow’s Milk Properties and Antibiotic Behaviour

Cow’s milk is complex fluid composed of lipids, proteins, carbohydrates and minerals in aqueous phase [30]. It typically requires sample preparation and extraction prior to analysis to avoid possible interference between fractions.

### 2.1. Raw Milk Composition

As shown in Table 1, water content is very high in milk. Lactose is the main carbohydrate contained in milk and plays a significant role in fermentation. It also regulates the water content of milk; thus, its content is not a variable but mostly constant in milk. Other carbohydrates present in raw milk are monosaccharides (such as glucose), sugar phosphates and oligosaccharides [31].

The fat content in cow’s milk can be divided into simple and complex lipids known as triglycerides and phospholipids, respectively [30]. The main simple lipid is triacylglycerol, and half of its content is synthesised by the mammary gland, and the other half is derived directly from the bloodstream [31]. Simple lipids comprise ~90% of the fat content in milk. On the other hand, phospholipids contain phosphorus and/or nitrogen and can form bridges between fatty and aqueous phases of milk due to their amphiphilic properties. Examples of milk-containing phospholipids are lecithin, phosphatidyl choline and phosphatidyl ethanolamine [38].

Dairy proteins consist of 3.5 g/100 g milk and can be divided into three classes as casein, whey and fat globule membrane protein, and they all contain vital amino acids [30]. Casein proteins have low solubility at pH 4.6, and the four major ones are known as α_s1_, α_s2_, β and *k* caseins. Whey proteins are soluble in a wide range of pH and are often referred to as serum proteins, with major ones being α-lactoglobulin and β-lactalbulin [39]. Fat globule membrane proteins are composed of both lipids and proteins and originate from the mammary gland epithelia [40]. They comprise only ~1–2% of the protein content of milk. A study analysing raw bovine milk had determined 20 different proteins categorised as the above classes and casein, whey and fat globule membrane proteins comprising of 80.4, 13.5 and 1.7 g/100 g protein, respectively [41].

### 2.2. Binding Properties of Antibiotics

Depending on their solubility properties, antibiotic molecules are not distributed evenly through the milk and essentially are binding to different fractions [32,35]. *Hydrophilic* antibiotics (such as some β-lactams, sulphonamides and fluoroquinolones) would concentrate in skim milk due to the absence of fat; hence, *lipophilic* drugs (such as some macrolides, e.g., tylosin) would tend to concentrate in, for instance, cream. In Table 1 some examples of antibiotic classes are linked to the respective fraction of milk they are likely to concentrate in, depending on their solubility properties. Generally, tetracyclines are lipophilic antibiotics, with the exception of oxytetracycline and chlortetracycline. A study proves that lipophilic tetracyclines tend to concentrate in the fatty fractions on milk rather than in aqueous such as skimmed milk [32]. Fat percentage in milk varies between cows on a farm. For example, cattle breed is one of the main factors for variation of fat content in cow’s milk. In particular, Jersey breed cows produce milk that is rich in lipids [31], suggesting that lipophilic tetracyclines would concentrate in Jersey cow’s milk after administration. 

Even though fats and carbohydrates comprise larger portions of bovine milk’s composition, proteins are much larger molecules that consist of hundreds of amino acid chains and fold into themselves. Table 2 below demonstrates the molecular weight’s significance of the major constituents of each fraction. The larger the molecule is, the larger the surface of binding it provides. 

In fact, not only lipophilicity but also protein-binding properties play a significant role in antibiotic concentration distribution in milk. Some antimicrobial drugs (such as tylosin) tend to concentrate in casein protein fraction, for example [32]. In terms of tetracyclines, their protein-binding properties vary, with doxycycline having the highest range of 82–93%. The hydrophilic oxytetracycline has the lowest range of 27–35%; tetracycline and chlortetracycline’s protein-binding properties range between 55–64% and 50–55%, respectively [37]. Hence, lipophilicity has some impact on the protein-binding properties of antibiotics and must be carefully considered prior to any analysis.

## 3. Chemical and Biological Properties of Tetracyclines

Tetracyclines are broad-spectrum antibiotics that are active against a range of Gram-positive and Gram-negative bacteria, as well as some intracellular bacteria, such as chlamydiae, mycoplasmas and rickettsiae, and protozoan parasites [48]. In particular, tetracyclines are used on cattle for the treatment of bovine respiratory diseases (BRD) and mastitis [49,50]. Tetracyclines can be sorted into three categories depending on the timeframe of their development. First-generation tetracyclines were discovered between 1948 and 1963, and second-generation tetracyclines in the period between 1965 and 1972 [16]. Third-generation tetracyclines were discovered in 1993 and are called glycylcyclines. These have the advantage of being active against first- and second-generation tetracycline-resistant organisms [51].

The most widely used tetracyclines in veterinary medicine are chlortetracycline (CTC), oxytetracycline (OTC), tetracycline (TC) and doxycycline (DC). With respect to the above three categories, CTC, OTC and TC are first-generation tetracyclines, whereas DC is a second-generation tetracycline. Glycylcyclines are not authorised for use on animals due to the lack of a maximum residue limit set for them [52].

The chemical structure of all tetracyclines consists of a tetracene (or naphthacene) core (Figure 2) and similar functional groups [53]. The addition of molecules such as (-Cl) or (-OH) determines the derived names of CTC or OTC, for instance. However, all tetracyclines contain a phenol group that makes them electroactive substances [54].

The chemical properties of CTC, OTC, TC and DC are summarised in Table 3 below.

### 3.1. Mode of Action

In cattle, tetracyclines are used to treat infections mainly caused by *Pasteurella multocida*, Mannheimia haemolytica and *Histophilus somni* [49]. This antibiotic class demonstrates a bacteriostatic activity; therefore, these antimicrobials inhibit the growth of bacteria [55]. Tetracyclines tend to bind mainly to the 30S ribosomal subunit with high affinity and prevent protein synthesis, as shown in Figure 3. However, this is reversible, and if the antibiotic is displaced, protein synthesis will continue within the bacterial cell [16,48]. 

### 3.2. Biotransformation, Excretion Routes and Withdrawal Time

Tetracyclines are generally not metabolised and are excreted from the body unchanged, with the exception of TC, 5% of which is metabolised to a less active metabolite called 4-epitetracycline [37,56]. All tetracyclines are excreted mainly via the liver (>50%) and kidneys (≤30%) or bind to tissues; however, 0.3% have been reported to be eliminated via the udder [14].

Withdrawal time can vary between antibiotic manufacturers, even with the same active ingredient ranging from 6 to 14 days. There are a couple of sprays used on cattle for dermatitis, for example, containing CTC or OTC that are reported to not have any effect on milk as per prescription. Their withdrawal time is said to be zero hours [57]. Terramycin is one of the most commonly used commercially available antibiotics containing the active ingredient of oxytetracycline. Its withdrawal period time for milk from cattle is listed to be 5 days [58]. 

### 3.3. MRL and Evidence of Presence in Milk

The maximum allowance for tetracyclines to be present solely or in a combination in milk is set at 100 µg/L in the EU and by the Food and Agricultural Organisation (FAO) and the WHO [11,12,59]. However, the US Food and Drug Administration (FDA) had set the limit of the sum of tetracyclines to be 300 µg/kg [60]. As mentioned earlier, some antibiotics are found to be stable in a wide range of temperatures, and tetracyclines are one of these antimicrobial agents [61]. A study examined tetracycline and oxytetracycline’s stability under pasteurisation (85°C) conditions, and both antibiotics were found to be relatively stable after processing with only 5.7% and 15.3% activity decrease, respectively [62]. The results of another study testing dairy products after pasteurisation and skimming process for traces of tetracyclines transferred from raw contaminated milk showed that the highest residues were found in curd, cheese and buttermilk with the highest residue values of 482 µg/kg, 561 µg/kg and 221 µg/kg, respectively [63].

Numerous researchers had tested raw or processed cow’s milk collected directly from farms or supermarkets and have detected tetracyclines above the allowed limit [3,64,65,66]. A couple of studies conducted in Iran, for example, had found tetracyclines in milk at a much higher concentration than 100 µg/L with a sample containing a concentration 2.5× higher than the allowed limit [64,67]. Another study completed in Turkey had analysed 100 raw milk samples for CTC, TC and OTC and found an average range of antimicrobials of 45.56–98.30 µg/L and the highest value of 135.32 µg/L of TC [68].

### 3.4. Persistence in the Environment

As mentioned in the introduction, one way of disposal of contaminated milk is by discarding it in a slurry tank or spraying it over the land on the farm. Tetracycline has been reported not only to be eliminated from the body in an active and unchanged form but was also found to persist and accumulate in the soil over a few months [69].

Oxytetracycline was found to persist in soil but with low mobility and therefore does not move easily to aquatic environments nearby [70]. The persistence in the soil was also reported by analysing soil for the presence of chlortetracycline, which was believed to be the most common tetracycline antibiotic found in the environment [71]. It is worth reporting that tetracyclines tend to form complexes with metal cations, particularly with calcium, magnesium, copper and chromium [72].

As discussed earlier, antimicrobial resistance is a major crisis occurring in the environment and in the human body. The first tetracycline-resistant bacteria strain was found in 1953 in *Shigella* sp. bacteria [16]. Since then, tetracycline resistance has been increasing and occurring in both human and animal bodies and in the environment due to constant exposure [73,74,75].

## 4. Existing Residue Test Methods

Two principal test methods are in use to ensure the safety standards of food, such as milk and meat, are met in terms of drug residues [76,77]. Existing on-site tests for drug residues in milk rely on one of two approaches. The first involves inhibition of a microorganism in the presence of antimicrobial residues [78], and the second involves antibodies immobilised on a surface, specific for the given residue, which can then be transduced into a human-readable signal [79]. These tests are useful for screening bulk milk on the farm before it is transferred into the tanker for transportation. Both approaches have merits from the perspective of biosensors and therefore are discussed in more detail below.

In addition to these practical, in-field screening techniques, more robust and accurate confirmatory methods (e.g., HPLC) are typically used to verify the bulk milk prior to offloading from the tanker and processing. One of the key challenges in the identification of antimicrobial residues relates to the fact that the further through the production chain contaminated milk travels, the greater the economic and environmental consequences because a greater quantity of milk must be discarded (Figure 4).

Figure 4 also marks potential checkpoints within the dairy chain where essentially biosensors could be implemented. As shown, in the absence of proper measures, the financial impact increases at each checkpoint within the chain, reaching huge financial losses and even litigation if contaminated milk is sold to customers.

### 4.1. Commercially Available Screening Tests

Commercial microbial inhibition and antibody tests are widely used within the dairy industry for testing milk at the early stages of the supply chain. These tests are effective for on-site testing and cost-effective when used in bulk milk, although a key disadvantage of both of them is that they are qualitative and rely upon the user to interpret the test result; in borderline cases, this could lead to a false-negative result. 

#### 4.1.1. Microbial Inhibition Test

Microbial inhibition tests rely on the use of well-characterised indicator organisms with known sensitivities to antibiotic drugs. *Geobacillus stearothermophilius,* for example, is used in the ISO standard tube diffusion test (Figure 5) [78]. Delvotest^®^ is the leading commercial microbial inhibition test with many variations. It has a relatively broad spectrum of antibiotic detection with a focus on β-lactams. Other antibiotic classes include macrolides, aminoglycosides, tetracyclines and sulfonamides. It is based on the growth of incubated bacteria within tubes or on a disk (in agar). As seen in Figure 5, when the sample is introduced in the tubes, if antibiotics are absent in the matrix, bacteria will grow, resulting in a drop in pH. This change in pH is then measured using an indicator dye within the growth media. If antibiotics are present in the sample, the indicator organism will not grow, and there will be no colour change in the tube. 

The limit of detection for tetracyclines ranges from 100 to >800 µg/kg. The main drawback of this test is that it requires long incubation periods and, therefore, cannot be used in real-time [80].

#### 4.1.2. Immune Receptor Tests

A commercial example of an immune receptor test is Charm^®^. It involves the preparation of an antibody–antibiotic complex as a ‘control’ and a free antibody on the surface. When the sample is introduced, the target antibiotic molecule binds to the free antibody and forms a ‘test’ antibody–antibiotic complex. Both ‘test’ and ‘control’ complexes then compete for the immune receptor, enzyme, for example. The result is based on the intensity of the ‘test’ and ‘control’ reactions. It is essentially a positive or negative result [81]. These tests are more expensive than microbial inhibition tests and require lab equipment and conditions when preparing the antibodies. They also have a very limited range of analytes, usually targeting β-lactam antibiotics, and due to their working principle, they are, therefore, unsuitable for real-time analysis in the field.

### 4.2. Laboratory-Based Analysis Techniques

More sensitive and laboratory-based techniques such as chromatography are also used for milk analysis. These are sometimes conducted at the processing plant as a confirmation before processing raw milk (red control dot in Figure 4). The analysis is generally carried out with high-performance liquid chromatography (HPLC). Other methods include thin-layer chromatography (TLC) and gas chromatography (GC). Protein extraction and sample clean-up of milk are required before analysis. This can involve strong chemicals and solid-phase extraction, which is expensive and time-consuming [61]. HPLC is also used for the determination of lactose content in dairy products after processing [82]. 

As mentioned, some antimicrobials have high protein-binding abilities; hence, the removal of proteins from the matrix can eliminate a portion of the antibiotics present in the sample. 

## 5. Emerging Electrochemical Biosensors for Tetracycline Residues

Electrochemical biosensors are attractive and widely used because they offer advantages of low overall cost, quicker assay time and opportunities for miniaturization of the platform [83,84]. A number of different strategies can be employed to electrochemically transduce a biorecognition reaction with tetracycline residues, as shown in Figure 6. Electrochemical analytical techniques broadly fall into one of three groups: potentiometric, amperometric and impedimetric [85]. In a typical electrochemical cell, three electrodes are used, although in some cases, two-electrode or four-electrode systems can be employed. A three-electrode cell consists of a working electrode (WE), a reference electrode (RE) and a counter electrode (CE). The WE (often platinum, gold or carbon) is generally modified with biorecognition elements to achieve specific and sensitive transduction of the biomarker of interest with the electrode. The CE and RE are used to provide a return path for the current generated at the WE and to control the potential of the WE, respectively. Measurement is then performed with an instrument called a potentiostat, which is capable of sensing the electrochemical properties of the WE.

Potentiometry is a well-known detection method based on the measurement of the potential at the surface of an electrode using a voltmeter [86,87]. In amperometric detection, a potential is applied to the WE, and oxidation or reduction reactions produce a measurable current [86,88]. Impedance spectroscopy is based on the perturbation of the WE with sinusoidal potentials of differing frequencies, which allow the conductance and capacitance of the electrochemical cell to be determined [89]. All three detection methods are employed in biosensors for the detection of tetracyclines, as described below. 

### 5.1. Tetracycline Detection Strategies

A number of detection strategies have been employed to electrochemically detect tetracycline residues. These can be broken down into six principal categories: enzyme-based, microbial inhibition, antibody-based, molecularly imprinted polymer-based, aptamer-based and direct detection. In this section, we explore each of these detection approaches in detail and give specific examples of developed biosensors in Table 4.

#### 5.1.1. Immunosensors

Immunosensors utilise the natural ability of antibodies to bind with high affinity to a target molecule or antigen (see Pollap et al. for a comprehensive review on the subject [116]). The binding affinity constant, expressed as Kd, of an antibody is a useful parameter for its performance evaluation in a sensing platform [117]. The higher the Kd value, the smaller amount of target species would be detected.

A competitive assay was developed by Conzuelo et al. for tetracycline and sulphonamide detection [93]. In this approach, the investigators immobilised polyclonal sheep antibodies onto the surface of the electrode and with Horseradish peroxidase (HRP) tagged tetracycline. HRP used H_2_O_2_ as the enzyme substrate and hydroquinone as a redox mediator to generate a measurable current. When unlabelled tetracycline residues are present in the sample, the HRP-tagged tetracycline is displaced from the electrode’s surface, and a measurable change in current is observed. Figure 7 is a representative diagram of this competitive electrochemical immunosensor and the chemistry on the surface of the electrode.

A biosensor developed by Que et al. using monoclonal rabbit antibodies had achieved very low detection limits of 0.006 µg/L [95]. They added biolabelled graphene sheets to the sample with platinum nanoparticles that catalysed a hydrogen evolution reaction (HER) and enhanced the signal.

The use of nanoparticles has also been reported by Liu et al. [96]. Magnetic nanoparticles modified with chitosan have also been used for simple and low-cost immunosensors with anti-TC monoclonal antibodies. MNPs offer good biocompatibility, low toxicity and high electron transfer. Chitosan films were used for their advantageous properties, including membrane forming ability, high mechanical strength and easy chemical modification. Detection limits as low as 0.03 µg/L were attained.

#### 5.1.2. Enzyme-Based Sensors

Enzyme-based sensors utilize specific enzymes in order to generate a bio-recognition reaction. The most well-known enzyme-based electrochemical biosensor is the blood glucose meter, which uses enzymes, such as glucose oxidase or glucose dehydrogenase, to measure glucose concentrations [118]. Besharati et al. used TetX2 monooxygenase for tetracycline detection achieving lower detection limits of 18 nM [91]. TetX2 monooxygenase is an enzyme isolated transposon from CTnDOT in *Bacteroides thetaoiotaomicron*. It contains flavin and requires NAD(P)H to catalyse the biotransformation of tetracycline to 11-a-thetaoiothaomicron [119]. Hence, the addition of NAD(P)H on the surface of the electrode resulted in its oxidation and improved response. However, the sensor is reported to require an hour from sample to result and wasn’t tested on raw milk samples [91]. Using the same enzyme, this sensor has been further developed with low limits of TC detection, which were confirmed by HPLC [90].

One of the drawbacks of enzyme-based sensors is that they can have very low specificity, leading to a false-positive result, particularly if molecules with similar chemical structures are present in the sample [120]. This might explain why few enzyme-based biosensors for the detection of tetracyclines have been reported.

#### 5.1.3. Microbial Inhibition Sensors

Microbial inhibition biosensors use the same principle as the tube diffusion test [78]; that is, the growth of sensitive microorganisms will be prevented or inhibited in the presence of the antimicrobial residue. The complex biochemistry of microorganisms means that a range of different parameters can be measured to determine inhibition, including pH and temperature changes [121]. Despite the opportunities, this provides for numerous electrochemical transduction strategies, few investigators have used this approach. Possible reasons for this are the long response times, the lack of quantitation of residue concentration and the non-specific nature of the approach for particular classes of residue. In one study, good sensitivity of tetracycline, oxytetracycline and chlortetracycline was found, but it took at least two hours for the sensor to detect the presence of residues [92].

#### 5.1.4. Direct Electrochemical Detection Techniques

Electrochemical sensors can be also developed without the use of biological material; hence, a direct detection on the surface of the electrode is achieved [122]. This technique relies upon the electroactive properties of the targeted antibiotics. For instance, tetracyclines contain redox-active groups in their chemical structure, allowing a direct electron transfer and producing a detectable electrochemical flow on the electrode’s surface [123]. Figure 8 is a generic schematic diagram of the direct detection of tetracycline on the surface of an electrode.

Screen-printed electrodes (SPEs) make attractive and affordable transducers for direct detection [124]. Modification of their surface, however, is typically required to enhance the electron transfer with the analyte. Often, such modification is achieved chemically using self-assembled monolayers (SAMs) that offer high molecular organisation and homogeneity [125]. Asadollahi-Baboli et al. report an example of modified SPE with SAM of cysteine on gold nanoparticles for simultaneous detection of tetracycline and cefixime (a cephalosporin) in milk [126]. Tetracycline was proved oxidizable; however, a tandem with chemometrics was required for optimal results. After careful consideration of the literature combined with previous work, the optimal pH of 2.9 was chosen for the buffer media. Empirically, it was discovered that the chosen pH provided the best results. The authors do not provide a further explanation for this. The redox properties of several organic molecules are known to change at different pH values, and presumably, a similar effect is at play with tetracycline at acidic pH, leading to the authors’ observations [127]. A prediction set, as well as a calibration set of solutions, were used to construct and evaluate the experimental design, which produced limits of detection of 0.52 µM for tetracycline in milk. Even though this state-of-the-art sensor does not fall within the requirements of LOD values below the stated MRLs for tetracycline, it is a good description of a possible way of detection of the redox-active analyte. Tetracycline behaviour in alkaline and neutral pH discussed by Anderson et al. [127].

Graphene-modified screen-printed carbon electrode was used for direct detection of tetracycline in milk by Filik et al., achieving a limit of detection down to 80 mM [128]. The milk samples, however, must be prepared and cleaned up by either solid-phase extraction or preliminary separation. Such sample preparation is not convenient practice in the field.

Direct electrochemical detection is generally not used for the analysis of complex matrices and milk as such. The lack of an immobilised bioreceptor reduces the specificity and the presence of other non-targeted, but redox-active compounds can only interfere with the signal and produce inaccurate results [122].

#### 5.1.5. Molecularly Imprinted Polymer (MIP) Sensors

MIP biosensors are based on the creation of synthetic materials containing specific receptor sites having a high affinity toward the target molecule. These are cross-linked organic structures containing pre-designed molecular recognition sites complementary in shape, size and functional groups to the template molecule [129]. A more detailed and comprehensive review of MIP biosensors was reported by Gui et al. [130]. A number of different approaches can be used to create MIPs, including the production of MIP particles for later integration into end use applications, such as sensors [131] and direct electropolymerisation onto the surface of a substrate [132].

MIPs have been used for the impedimetric detection of doxycycline [100]. *O*-Phenylenediamine was electropolymerized on the surface of a multi-walled glassy carbon nanotubes electrode. The use of carbon nanotubes provided a larger surface area and better conductivity, whilst the MIP layer provided the selectivity of the sensor to doxycycline when compared to analysis of other antibiotics. This sensor was applied to human serum samples for DC detection.

A biosensor was developed using polymerised microporous metal-organic framework directly onto the gold electrode surface, and p-aminothiophenol was electropolymerized [98]. Extremely low detection limits of 0.22 fM were achieved in honey in the recovery range of 101.8%−106.0%. Gold nanoparticles were used as a signal amplifier due to their small size, great stability, great conductivity and good catalytic activity.

#### 5.1.6. Aptasensors

Aptamers are fragments of DNA that fold in a predictable manner to create a ligand with a high affinity for a biological target molecule [133]. They can be also classified as ‘chemical antibodies’ and are attractive candidates for biosensors because they can be synthetically produced once a candidate has been identified. Aptamer candidate sequences are identified through the systematic evolution of ligands by the exponential enrichment (SELEX) process [134]. The SELEX process enables the fabrication of aptamers also for non-immunogenic and toxic targets, such as antimicrobial residues. They are also stable and are not affected by temperature; however, for optimal performance, the pH and ionic strength of the electrolyte solution must be constant [135,136]. A comprehensive review by Li et al. focuses on electrochemical biosensors using aptamers for analysis of food and water samples [137]. Figure 9 is a generic schematic diagram of the detection method of an aptasensor.

A classic example of an electrochemical sensor for tetracycline detection developed using aptamers as the bioreceptor was reported by Kim et al. [107]. The biotin labelled aptamer sequence was immobilized on a gold WE, and it was selected to specifically bind to tetracycline. The CE was also made of gold with a silver RE. The aptasensor showed good selectivity to TC. This was demonstrated when a mixture of TC:OTC:DC was also introduced to the sensor, and it picked up only the TC, which had proved the high specificity of the selected aptamer. The limit of detection of this aptasensor was 10 nM which is much lower than the MRL for TC, mentioned in the previous section.

An innovative approach was reported by Rad et al. combining tetracycline-sequenced aptamers and molecularly imprinted polymers, immobilised one after another on the electrode surface [99]. The working electrode material of choice was glassy carbon with gold nanoparticles. Dopamine was used as an additional ‘seal’ on top of the aptamer. Then, tetracycline was extracted from the aptamer sequence on the electrode surface, leaving empty cavities that are to be filled by potentially present TC in the introduced sample. The new TC molecules falling into the cavities essentially produce a charge on the redox probe, which is measured at a peak voltage of 0.22 V and thus achieving a limit of detection as low as 144 fM.

The detection of OTC with a two-electrode system using aptamer-based single-walled carbon nanotubes has also been carried out by Yildirim-Tirgil et al. [110]. The investigators had achieved not only very low detection limits and an extremely fast response time of only 10 min but also good reproducibility, suggesting that this approach could form the basis of a commercial solution.

Multi-walled carbon nanotubes (MWCNTs) have been widely used as the basis for a number of biosensors for the detection of tetracycline. MWCNTs on the surface of an electrode had been used as an amplifier in a study by Zhou et al., providing a larger and denser surface for the aptamers to attach [111]. Acid treatment was applied to MWCNTs before coating on the glassy carbon electrode by providing an excellent electrical conductivity and enhancing the charge transport. This amplification had resulted in higher sensitivity and great selectivity of the biosensor when it was compared to the results from a performance without the MWCNT coating.

### 5.2. Limitations of Existing Sensors and Future Needs

As discussed, biosensors offer promising alternatives to current antimicrobial residue testing methods for milk. However, there are limitations to existing platforms that should be a priority for future researchers in this area to facilitate full-time manufacturing and use.

#### 5.2.1. Specificity and Sensitivity

Firstly, sensor specificity is essential to avoid unacceptable levels of false-positive results [139]. In this review, different types of bioreceptors were discussed as well as direct detection of tetracyclines on a modified electrode surface. False-positive results can be a factor derived not only from a direct detection method but also from the similarity of the chemical structures of several antibiotics from the same class, such as tetracyclines. Hence, the use of a highly specific biological material must be carefully considered prior to immobilisation.

Another parameter used for evaluation is the sensitivity of a platform. The set MRL values for all antibiotics are guidance for target limits of detection; however, many of the biosensor strategies discussed in this paper do not meet the current threshold levels. This may not be a barrier to sensors that are used to detect residues in a milk sample from a single cow but means that sensitivity must improve for applications where testing of milk prior to consumption is required.

In recent years, working electrode surface modification with nanomaterials had become a widely-used technique for signal amplification. The use of nanomaterials in the reported examples of electrochemical biosensors in Table 4 was highlighted where applicable. Materials such as gold nanoparticles, graphene oxide and carbon nanotubes are attractive options due to their advantageous mechanical and electrical properties. These are discussed in more depth in a review by Alsaiari et al. [140].

#### 5.2.2. Sensor Cost

Biosensor cost is also an important factor both in terms of the design development cost and also the recurring consumable cost. Electrochemical devices are advantageous because production costs can be very low, but this is also dependent upon scales of economy and market size. Therefore, biosensor strategies that are adaptable to different applications are more attractive so that scales of the economy can be achieved.

#### 5.2.3. Sample Handling

Electrochemical detection offers the ability to shrink the sensing platform to make it easier to operate as well as enabling multi-sites for analysis. Microfluidic platforms are an innovative approach to tackle this challenge and offer a point-of-care monitoring system not only for environmental but also for clinical and medical samples. A developed electrochemical platform by Kling et al. had introduced eight sites for analysis, and the required sample volume was reduced drastically [141]. Tetracycline and pristinamycin were detected simultaneously in human plasma, achieving very low limits of detection of 6.33 and 9.22 ng/mL, respectively. A comprehensive review of microfluidic devices for milk analysis was reported by Ng et al. [142].

A future perspective is the fabrication of a multi-channel microfluidic platform that might be implemented in the milking machinery to monitor antibiotic residues directly before milk reaches the bulk tank. The challenges with its application in real-life conditions may, however, include biofouling on the electrode surface. This might interfere with the immobilised molecules and disrupt the biorecognition reaction, hence, result in false-positive or false-negative values. Antifouling strategies were discussed in a thorough review by Lin et al. [143].

## 6. Conclusions

When treated with antibiotic drugs, cows secrete a proportion of these as residues in milk, rendering it unfit for human consumption. Maximum residue limits are set globally to maintain the presence of antibiotics in food and avoid human health risks such as possible allergic reactions. To avoid health, environmental and financial risks imposed by this, the dairy industry is obliged to perform tests on raw milk within the supply chain. Here, we reviewed the methods currently employed by the dairy industry, which include screening and confirmatory tests and highlighted the limitations of existing commercially available testing approaches. Electrochemical biosensors are then outlined as an alternative for low-cost, rapid detection and could be suitable for real-time analysis. In particular, the electrochemical detection method is an attractive choice due to its high specificity and low-cost manufacturing. In recent years, the development of electrochemical biosensors has expanded and improved with the synthesis of aptamers; however, further research is still required to support the detection of the complete family of tetracyclines and also for the concurrent detection of multiple different residues.

## Figures and Tables

**Figure 1 biosensors-11-00232-f001:**
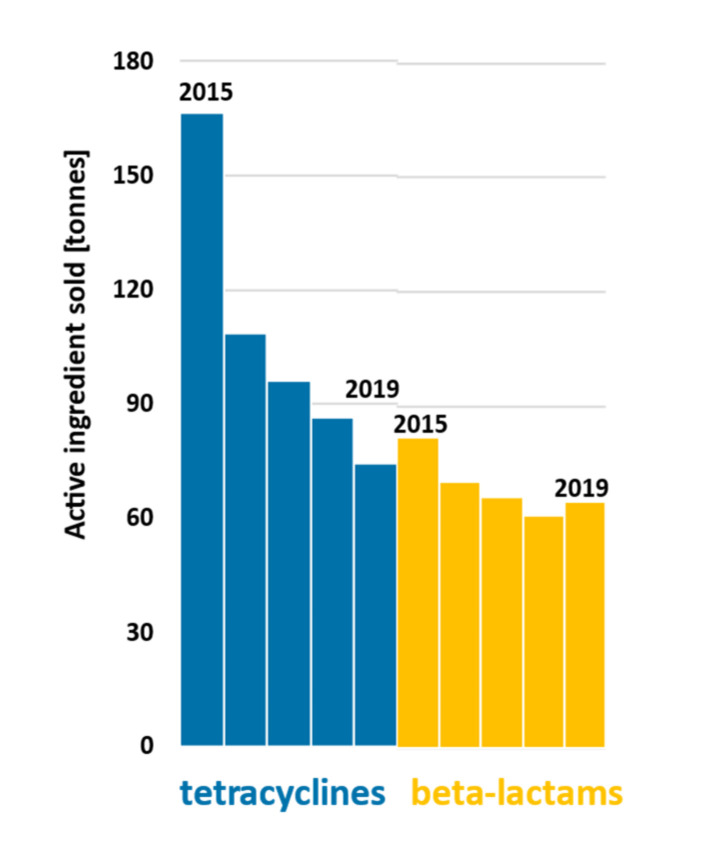
Sales of tetracyclines and beta-lactams between 2015 and 2019. (Data taken from UK-VARSS Report, 2019, Table 1.3) [17].

**Figure 2 biosensors-11-00232-f002:**
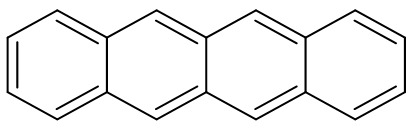
The chemical structure of tetracene (or naphthacene)—the core of all tetracyclines.

**Figure 3 biosensors-11-00232-f003:**
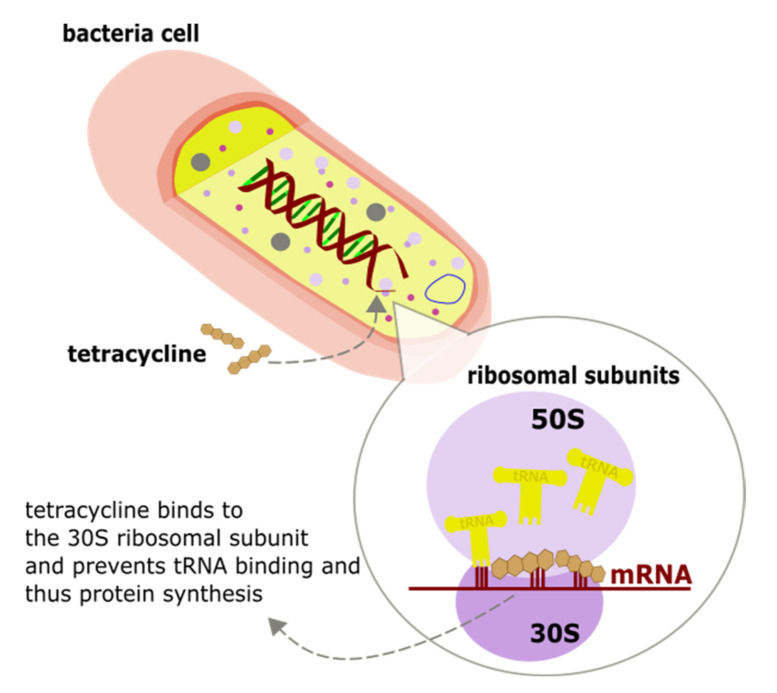
The mode of action of tetracyclines; bacterial cell inhibition by binding to the 30S ribosomal subunit and protein synthesis prevention.

**Figure 4 biosensors-11-00232-f004:**
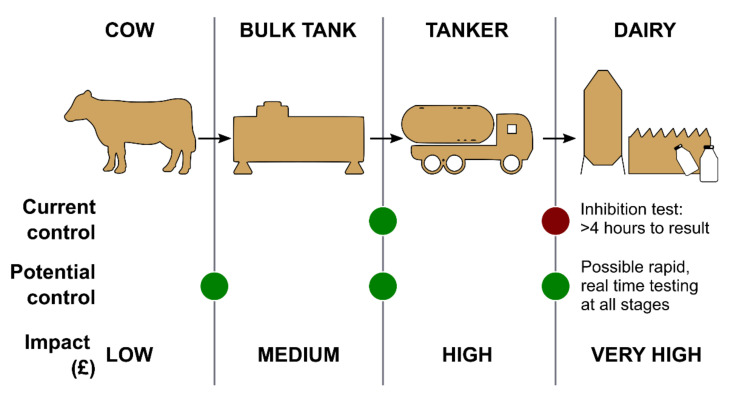
Current and potential testing checkpoints within the dairy supply chain, where financial impact increases with each step of testing towards the retailer.

**Figure 5 biosensors-11-00232-f005:**
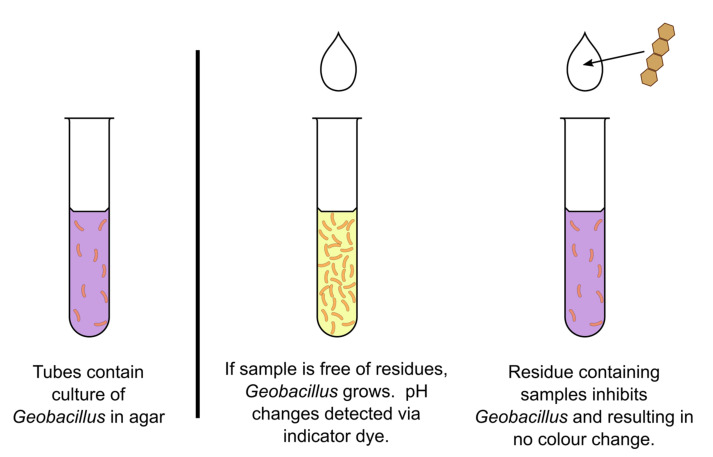
A schematic diagram of the microbial inhibition test for the detection of antimicrobial residues in milk.

**Figure 6 biosensors-11-00232-f006:**
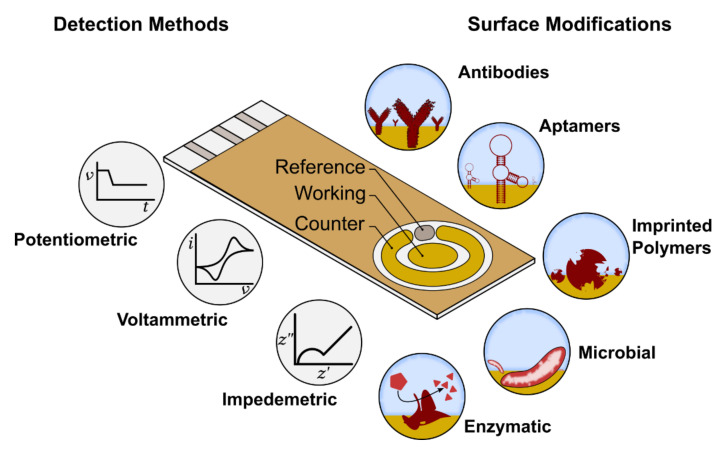
Electrochemical detection in a biosensor, where reference, working and counter respond to electrodes, which are arranged in a manner allowing the sample to be in contact with all three at the same time. The range of surface modifications is performed on the working electrode, which acts as the transducer in the biorecognition reaction.

**Figure 7 biosensors-11-00232-f007:**
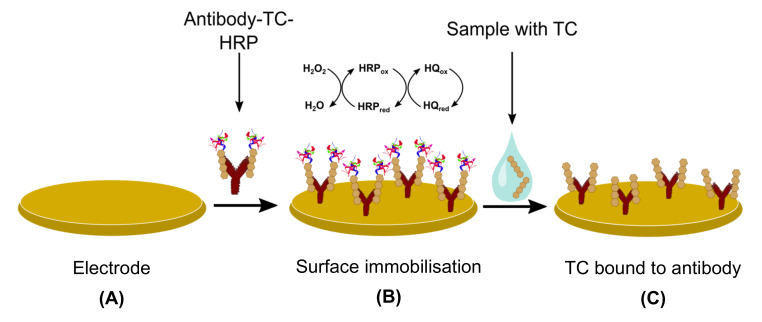
A schematic example of a competitive electrochemical immunosensor for the detection of tetracycline (TC). In this example, HRP is used to amplify the signal and identify the presence of TC. (**A**) The gold electrode surface is cleaned ready to receive a (**B**) TC antibody. After immobilisation on the surface, HRP tagged TC is added, which generates H_2_O_2_ in the presence of hydroquinone. Several strategies exist for the detection of H_2_O_2_ at an electrode surface. (**C**) When a sample is added containing TC, the HRP–TC complex is displaced from the antibody, and a drop in H_2_O_2_ production is electrochemically observed. Adapted from Conzuelo et al. [93].

**Figure 8 biosensors-11-00232-f008:**
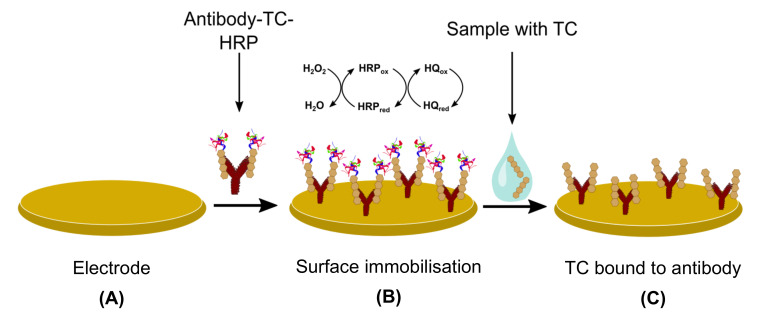
A schematic example of direct detection of tetracycline (TC). At high oxidation potentials, TC can be directly oxidized without dependence upon an affinity agent. After cleaning the electrode surface (**A**), an anti-fouling passivation layer is added to the electrode to create an alkanethiol layer (**B**). (**C**) at high oxidation potentials (0.85 v vs Ag/AgCl), TC is oxidised, resulting in a Faradaic charge transfer proportional to the concentration of TC present.

**Figure 9 biosensors-11-00232-f009:**
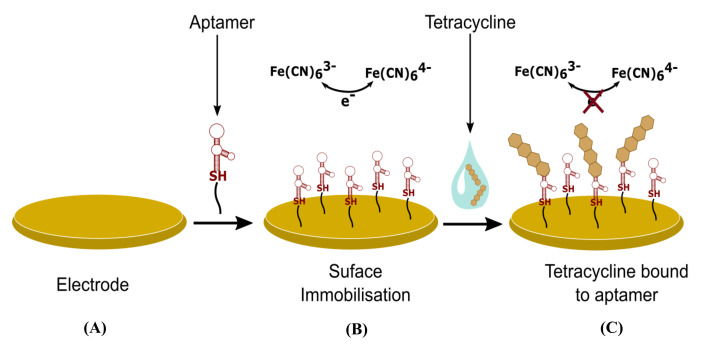
A schematic example of aptamer sensor for detection of TC. (**A**) The gold electrode surface is cleaned to remove contaminants. (**B**) Aptamer containing a thiol (SH) linker is immobilised to the electrode surface. At this stage, an alkanethiol passivation layer is also often added to the surface. A redox couple such as FeCN(6)^3-^/^4-^ can then be used to test for changes in the electrode surface. (**C**) When TC binds to the surface, a change in the Faradaic current is observed through impedance spectroscopy or voltammetric detection techniques. Adapted from Mehennaoui et al. [138].

**Table 1 biosensors-11-00232-t001:** Cow’s raw milk composition (g/100 g milk) and the main constituents of each fraction; depending on their solubility properties, different antibiotics bind to certain fractions of milk, examples listed.

Fraction of Milk	Content (g/100 g Milk)	Main Constituents	Antibiotics Likely to Concentrate	Reference
Water	87.2	-	oxytetracycline	[30,32,33]
			chlortetracycline	
			benzylpenicillin ^1^	
			amoxicillin ^2^	
			sulfadimethoxine ^3^	
			ciprofloxacin ^4^	
Carbohydrates	4.9	Lactose	-	[30]
		Monosaccharide		
		Sugar phosphates		
		Oligosaccharides		
Fats	3.7	Triglycerides Phospholipids	tetracycline	[30,32,34,35,36]
			doxycycline	
			tylosin ^5^	
			aminoglycosides	
			thiamphenicol	
Proteins	3.5	Casein Whey Fat globule membrane	chlortetracycline	[30,32,35,37]
			tetracycline	
			doxycycline	
			tylosin ^5^	
Minerals	0.72	Ca, Na, Mg, P, Cl, K	-	[30]

^1^ first-generation penicillin; ^2^ second-generation penicillin; ^3^ sulphonamide antibiotic class; ^4^ fluoroquinolone antibiotic class; ^5^ macrolide antibiotic class.

**Table 2 biosensors-11-00232-t002:** Molecular weights of major constituents composing raw cow’s milk fractions; Milk fractions being carbohydrates (lactose, glucose), fats (triacylglycerol, lecithin) and proteins (k casein, α-lactoglobulin).

Constituent	Molecular Weight (g/mol)	Reference
Lactose	342.3	[42]
Glucose	180.2	[43]
Triacylglycerol	855.4	[44]
Lecithin	758.1	[45]
*k* casein	18 974	[46]
α-lactoglobulin	14 178	[47]

**Table 3 biosensors-11-00232-t003:** The properties of commonly used tetracyclines in veterinary medicine (chemical data taken from PubChem).

Active Ingredient	Molecular Formula	Molecular Weight (g/mol)	Solubility (mg/L) at 25 °C
Chlortetracycline	C_22_H_23_ClN_2_O_8_	478.9	~500
Oxytetracycline	C_22_H_24_N_2_O_9_	460.4	312
Tetracycline	C_22_H_24_N_2_O_8_	444.4	231
Doxycycline	C_22_H_24_N_2_O_8_	444.4	50 000

**Table 4 biosensors-11-00232-t004:** Examples of electrochemical biosensors for tetracycline residues detection defined by main parameters used for the description of any biosensor.

Bioreceptor	Working Electrode Material	Selectivity	Detection Method	Limit of Detection	Linearity Range	Response Time	Reference
Enzyme-based sensors							
NAD(P)H-dependent TetX2 enzyme	Nano-porous glassy carbon	TC	amperometric	40 nM	0.1–0.8 µM	-	[90]
TetX2 monooxygenase enzyme	Glassy carbon	TC	amperometric	18 nM	0.5–5 µM	≤60 min	[91]
Microbial sensors							
*Escherichia coli*	RE and CE galvanic cell	TC, OTC, CTC	potentiometric	≤25 µg/L	-	120 min	[92]
Immunosensors							
Anti-TC polyclonal sheep antibody on protein-G-MBs	Dual screen-printed carbon	TC	amperometric	0.858 µg/L	10^−3^–10^−4^ µg/L	30 min	[93]
Anti-TC polyclonal sheep antibody on protein-G-MBs	Screen-printed carbon	TC OTC CTC DC	amperometric	8.9 µg/L 1.2 µg/L 66.8 µg/L 0.7 µg/L	17.8–189.6 µg/L 4.0 #x2013;242.3 µg/L 144.2–2001.9 µg/L 2.6–234.9 µg/L	30 min	[94]
Anti-TC monoclonal rabbit antibody	* Gold modified with PtGN	TC	amperometric	0.006 µg/L	0.05–100 µg/L	-	[95]
Anti-TC monoclonal antibody	* Gold modified with MNPs	TC	amperometric	0.0321 µg/L	0.08–0.1 µg/L	20 min	[96]
Molecularly Imprinted Polymer sensors							
MAA-AIBN MIP	Pt/Ti	TC	amperometric	26 µg/L	0.1–10 mg/L	-	[97]
* AuNPs added to p-aminothiophenol MIP	MMOF modified gold	TC	potentiometric	0.22 fM	224 fM–22.4 nM	30 min	[98]
Dopamine MIP and TC-aptamer	* Glassy carbon modified with AuNPs	TC	impedimetric	144 fM	0.5–100 nM	45 min	[99]
o-Phenylenediamine MIPs	* MWGCNTs	DC	amperometric	1.3 × 10^−2^ μM	0.05–0.5 μM	15 min	[100]
MIP pyrrole	* Screen printed carbon modified with AuNPs	TC	potentiometric	0.65 µM	1–20 µM	-	[101]
MAA-MIPs	* MWGCNTs modified with AuNPs	TC	amperometric	0.04 mg/L	0.1–40 mg/L	-	[102]
Aptasensors							
39-mer thiolated TC-binding aptamer	* Pencil graphite modified with AuNPs/RGO	TC	impedimetric	3 × 10^−17^ M	10^−16^–10^−6^ M	90 min	[103]
M-shaped aptamer (Apt-CSs) with Exo I	Gold	TC	amperometric	0.74 nM	1.5 nM–3.5 µM	-	[104]
TC-aptamer	Glassy carbon	TC, OTC	amperometric	1 µg/L	0.1–100 µg/L	5 min	[105]
TC-aptamer	Gold	TC	impedimetric	10 µg/L	10–3000 µg/L	15 min	[106]
ssDNA aptamer	Screen-printed gold	TC	amperometric	10 nM	10 nM–10 µM	-	[107]
β-cyclodextrin-aptamer	Gold	TC	impedimetric	0.008 nM	0.01–100 nM	-	[108]
Amino modified aptamer	Glassy carbon	OTC	impedimetric	2.29 × 10^−10^ g/mL	10^−9^–10^−4^ g/mL	60 min	[109]
ssDNA aptamer	Single walled carbon nanotube	OTC	amperometric	1.125 µg/L	10–75 µg/L	10 min	[110]
TC-aptamer and multi-walled carbon nanotubes	* Glassy carbon modified with MWCNTs	TC	amperometric	5 nM	10^−^^8^–10^−^^5^ M	30 min	[111]
TC-aptamer	* Glassy carbon modified with PB-CS-GA system and AuNPs	TC	amperometric	0.32 nM	10^−^^9^–10^−^^2^ M	-	[112]
TC-aptamer	* Glassy carbon modified with graphene oxide nanosheets	TC	impedimetric	29 fM	0.1 pM–10 µM	50 min	[113]
TC-aptamer	* rGo-Fe_3_O_4_/sodium alginate modified screen-printed carbon	TC	amperometric	0.6 nM	1 nM–5 µM	-	[114]
TC-aptamer and bio-cDNA aptamer	* Glassy carbon modified with MoS_2_-TiO_2_@Au composite	TC	amperometric	0.05 nM	0.15 nM–6.0 × 10^−^^6^ M	80 min	[115]

* Nanomaterials used for signal amplification. Note: TC = tetracycline; OTC = oxytetracycline; CTC = chlortetracycline; DC = doxycycline; RE = reference electrode; CE = counter electrode; AuNPs = gold nanoparticles; MBs = magnetic beams; PtGN = platinium graphene nanosheets; MNPs = metal nanoparticles; MMOF = microporous metal organic framework; MAA-AIBN = methacyclic acid-2-2′-azobisisobutyronitrile; MWGCNTs = Multi-walled glassy carbon nanotubes; RGO = reduced graphene oxide; CSs = complimentary strands; PB-CS-GA = Prussian blue-chitosan-glutaraldehyde; rGo-Fe_3_O_4_ = reduced graphene oxide-magnetite; bio-cDNA = biotin complementary DNA; MWCNTs = multi-walled carbon nanotubes.

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
