# Peer review of "Emerging Electrochemical Sensors for Real-Time Detection of Tetracyclines in Milk"

_biosensors, 2021, doi:10.3390/bios11070232_

Round 1

Reviewer 1 Report

Comments:
In this work, the authors presented a review on biosensor detection strategies for tetracyclines. The review will be helpful for the potential readers; however, the article has some questions.

  1. This review based on only electrochemical detection strategies of Hence, the authors should modify the title with more specific or relevant of the article. For example, “Emerging electrochemical sensors for real-time detection of tetracyclines in milk” look suitable for this article.
  2. The authors should discuss the drawbacks of current detection techniques and future perspectives.
  3. A detailed discussion on nanomaterials-based electrochemical tetracycline detection is highly recommended to make this article more attractive. If possible, a similar table like Table 4 can be presented for nanomaterials based on electrochemical tetracycline detection.
  4. It would be better to discuss possible challenges associated with real-time electrochemical detection of tetracycline. For example, possibilities of false-positive results, the effect of electrode surface in real-life applications, economic feasibility, and so on.
  5. Microfluidic-based electrochemical tetracycline detection should be discussed if possible.

Reviewer 2 Report

  1. It is worth explaining the mechanism of tetracycline on the inhibition of bacteria growth. The author may want to add a schematic graph.
  2. Line 289, the author claimed commercial microbial inhibition is rapid, which is not consistent with the later statement. Actually, microbial inhibition is time-consuming.
  3. In Figure 5, the Amperometric inset figure is not correct. Amperometric measures the current over time. The inset figure is cyclic voltammetry.
  4. Line 345. It should be "control the potential of the WE respectively."  
  5. Line 447, the author gave an example of an electrochemical biosensor with a limit of detection around 0.52 uM, which is around 231 ug/L. In Section 3.3, the author said, "The maximum allowance of tetracyclines to be present solely or in a combination in milk is set at 100 ug/L". Thus, many of the investigated sensors with LODs exceeding this value are considered to be ineffective for the determination of tetracyclines. The sensitivity of the method must coincide with relevant expectations for tetracyclines in the milk samples. Please explain it.
  6. It is necessary to provide enough data figures from referred publications to make the readers easy to follow. For example, the cyclic voltammograms of tetracycline. Please add 1-2 figures for each technique (Immuno sensors, enzyme-based sensors, electrochemical detection techniques, etc.)
  7. Line 445, the author mentioned 2.9 as the optimized pH value. This is due to the pKa of tetracycline. The author needs to explain in more detail. How does the pH value affect the sensitivity of the sensor?
  8. Line 483, the author said the aptamer is not affected by temperature or pH changes. This statement is wrong. Actually, the affinity of aptamer toward the target molecule is sensitive to the pH change. Please check this article: DOI: 10.1016/j.bioelechem.2006.03.012.

Round 2

Reviewer 2 Report

Congratulations to the author. I have no further questions.